# The Metabolism of Epoxyeicosatrienoic Acids by Soluble Epoxide Hydrolase Is Protective against the Development of Vascular Calcification

**DOI:** 10.3390/ijms21124313

**Published:** 2020-06-17

**Authors:** Olivier Varennes, Romuald Mentaverri, Thomas Duflot, Gilles Kauffenstein, Thibaut Objois, Gaëlle Lenglet, Carine Avondo, Christophe Morisseau, Michel Brazier, Saïd Kamel, Isabelle Six, Jeremy Bellien

**Affiliations:** 1MP3CV, EA7517, CURS (Centre de Recherche Universitaire en Santé), University of Picardie Jules Verne, 80025 Amiens, France; olivier.varennes@gmail.com (O.V.); romuald.mentaverri@u-picardie.fr (R.M.); thibaut.objois@gmail.com (T.O.); gaelle.lenglet@u-picardie.fr (G.L.); carine.avondo@u-picardie.fr (C.A.); michel.brazier@u-picardie.fr (M.B.); said.kamel@u-picardie.fr (S.K.); 2Department of Biochemistry, Amiens-Picardie University Hospital, 80054 Amiens, France; 3Department of Pharmacology, Rouen University Hospital, CEDEX 1, 76031 Rouen, France; thomas.duflot@chu-rouen.fr; 4INSERM U1096, Normandy University, UNIROUEN, F-76000 Rouen, France; 5INSERM U1260, University of Strasbourg, F-67000 Strasbourg, France; gilles.kauffenstein@gmail.com; 6Department of Entomology and Nematology and UCD Comprehensive Cancer Center, University of California, Davis, CA 95616, USA; chmorisseau@ucdavis.edu

**Keywords:** vascular calcification, soluble epoxide hydrolase, phosphatase, epoxyeicosatrienoic acids

## Abstract

This study addressed the hypothesis that soluble epoxide hydrolase (sEH), which metabolizes endothelium-derived epoxyeicosatrienoic acids, plays a role in vascular calcification. The sEH inhibitor *trans*-4-(4-(3-adamantan-1-yl-ureido)-cyclohexyloxy)-benzoic acid (*t*-AUCB) potentiated the increase in calcium deposition of rat aortic rings cultured in high-phosphate conditions. This was associated with increased tissue-nonspecific alkaline phosphatase activity and mRNA expression level of the osteochondrogenic marker Runx2. The procalcifying effect of *t*-AUCB was prevented by mechanical aortic deendothelialization or inhibition of the production and action of epoxyeicosatrienoic acids using the cytochrome P450 inhibitor fluconazole and the antagonist 14,15-epoxyeicosa-5(Z)-enoic acid (14,15-EEZE), respectively. Similarly, exogenous epoxyeicosatrienoic acids potentiated the calcification of rat aortic rings through a protein kinase A (PKA)-dependent mechanism and of human aortic vascular smooth muscle cells when sEH was inhibited by *t*-AUCB. Finally, a global gene expression profiling analysis revealed that the mRNA expression level of sEH was decreased in human carotid calcified plaques compared to adjacent lesion-free sites and was inversely correlated with Runx2 expression. These results show that sEH hydrolase plays a protective role against vascular calcification by reducing the bioavailability of epoxyeicosatrienoic acids.

## 1. Introduction

Vascular calcification is a highly regulated process observed during aging, hypertension, diabetes, and particularly chronic kidney disease (CKD). It contributes to elevated risk of cardiovascular events and mortality [1,2,3,4]. Dysregulated mineral metabolism characterized by the long-term elevation of serum phosphate is a main driver of vascular calcification, by promoting osteochondrogenic differentiation of vascular smooth muscle cells (VSMCs) [1,2,3,4]. Currently, there are very limited therapeutic options for either the prevention or treatment of vascular calcification, and new pharmacological targets are needed.

Increasing experimental evidence shows that the endothelium not only is a source of osteoprogenitor cells but also regulates vascular calcification through the release of signaling molecules such as nitric oxide (NO), which prevents VSMC osteochondrogenic differentiation [5,6,7]. Endothelial cells produce also epoxyeicosatrienoic acids (EETs), which are formed by the action of cytochrome P450 epoxygenases on arachidonic acid and display vasodilator and anti-inflammatory actions [8,9]. EETs are rapidly converted by soluble epoxide hydrolase (sEH), located in endothelial cells and VSMCs, to the biologically less active dihydroxyeicosatrienoic acids (DHETs), and inhibitors of sEH have been successfully developed to potentiate EET bioavailability [8,9]. Interestingly, two clinical studies evidenced arterial calcification in subjects carrying the R287Q single nucleotide polymorphism of the sEH gene *EPHX2*, which induces a decrease in sEH activity in vitro, suggesting a deleterious impact of EETs on vascular calcification [10,11,12]. In contrast, based on the measurements of plasma DHET levels, a recent clinical study suggested that EETs may be protective against the development of vascular calcification [13].

The aim of the present study was to assess the role of EETs and sEH in the vascular calcification process using mineralization assays of rat aortas and human VSMCs, and complementary experiments on human calcified carotid arterial materials were performed.

## 2. Results

### 2.1. High-Phosphate Conditions Promote Vascular Calcification

To evaluate the impact of sEH hydrolase and phosphatase inhibitors ex vivo on vascular calcification, we cultured isolated rat aortas in normal and high-phosphate conditions as previously described [14,15,16]. We confirmed a time- and dose-dependent effect of high-phosphate conditions on rat aortic calcification [16], as demonstrated by calcium deposition (Figure 1A,B) and as illustrated by Alizarin red and Von Kossa staining of calcium accumulation in aortas cultured in 3.8 mM Pi (Figure 1C) without change in aortic viability (Figure 1D). Moreover, high-phosphate conditions induced a decreased mRNA expression level of the contractile marker smooth muscle myosin heavy chain (SMMHC) and increased mRNA levels of the osteochondrogenic transcription factors runt-related transcription factor 2 (Runx2), Msh homeobox 2 (Msx2) and sex-determining region Y-box 9 (Sox9) (Figure 1E). In addition, there was an increase in tissue-nonspecific alkaline phosphatase (TNAP) activity (Figure 1F) and a decrease in pyrophosphate (PPi) level (Figure 1G), an endogenous inhibitor of ectopic calcification which is hydrolyzed by TNAP [16,17,18], in the culture supernatants of high-phosphate-cultured aortic rings.

### 2.2. Inhibition of sEH Hydrolase Enhances Vascular Calcification

High-phosphate conditions did not significantly affect aortic sEH mRNA expression level (Figure 1H) or the ratio of the preferential substrate of the sEH hydrolase domain 14,15-EET to its metabolite 14,15-DHET (Figure 2A) quantified in the aortic culture supernatant [19,20], arguing against a modification in sEH hydrolase activity. However, we observed that *trans*-4-(4-(3-adamantan-1-yl-ureido)-cyclohexyloxy)-benzoic acid (*t*-AUCB), a potent and specific inhibitor of sEH hydrolase [21,22], induced a dose-dependent increase in the calcium content of aortic rings cultured in 3.8 mM Pi (Figure 2B) but not of aortic rings cultured in 0.9 mM Pi (Appendix A
Appendix A). Since a significant potentiation of aortic calcification was reached at 10 µM, as illustrated by Alizarin red and Von Kossa staining (Figure 2C) without affecting aortic viability (Appendix A
Appendix A), the following experiments were performed at this concentration. Efficient inhibition of the hydrolase domain of sEH with 10 µM *t*-AUCB was confirmed by the decrease in the 14,15-DHET-to-14,15-EET ratio (Figure 2A). We measured no significant changes in the mRNA levels of SMMHC, Msx2 or Sox9; however, *t*-AUCB an increased Runx2 mRNA level compared to 3.8 mM Pi alone (Figure 2D). In addition, TNAP activity was increased by *t*-AUCB (Figure 2E) but we could not detect a further decrease in the PPi level (Figure 2F).

### 2.3. Role of the Endothelium in the Potentiation of Vascular Calcification by sEH Hydrolase Inhibition

We then assessed the role of endothelium-derived EETs in the effects of the sEH hydrolase inhibitor *t*-AUCB on aortic calcification. First, the mechanical deendothelialization of aortic rings [15], which was confirmed by the near-complete abolition of CD31 staining (Figure 3A), increased the aortic calcium deposition in response to 3.8 mM Pi compared to non-deendothelialized rings (Figure 3B), emphasizing a protective role of the endothelium against vascular calcification. In addition, 10 µM *t*-AUCB no longer modified the calcium content of aortic rings cultured in Pi 3.8 mM when the endothelium was removed (Figure 3B). The importance of the endothelium was confirmed in human VSMCs that express sEH protein (Figure 4A). In fact, 10 µM *t*-AUCB did not affect the calcification of VSMCs (Figure 4B) cultured for 14 days in high-phosphate conditions, as previously described and validated [15,23,24], without change in VSMC viability (Figure 4C).

### 2.4. Role of EETs in the Potentiation of Vascular Calcification Induced by sEH Hydrolase Inhibition

We then assessed whether the inhibition of the degradation of endothelium-derived EETs by sEH hydrolase contributes to the procalcifying effect of *t*-AUCB. The inhibition of EET synthesis using the cytochrome P450 inhibitor fluconazole (100 µM) [22] prevented a decrease in the 14,15-DHET-to-14,15-EET ratio induced by *t*-AUCB (Figure 2A), and this was associated with a prevention of the *t*-AUCB procalcifying effect (Figure 2B,C) without change in aortic viability (Appendix A
Appendix A). Fluconazole did not modify the mRNA levels of SMMHC, Msx2 and Sox9 but prevented the increase in Runx2 mRNA level (Figure 2D) and in TNAP activity induced by *t*-AUCB (Figure 2E) as well as markedly increased PPi level (Figure 2F).

Because cytochrome P450 enzymes blocked by fluconazole produce not only EETs but also other epoxides from various omega-3 and omega-6 fatty acids [9], we performed additional independent experiments using the EET antagonist 14,15-epoxyeicosa-5(Z)-enoic acid (14,15-EEZE: 1 µM) [25]. The addition of 14,15-EEZE alone did not change the calcium content of aortic rings cultured in 3.8 mM Pi but prevented the increase induced by 10 µM *t*-AUCB (Figure 5A). Moreover, direct addition of 11,12-EET and 14,15-EET at 1 µM in the culture medium of aortic rings mimicked the effects of *t*-AUCB with a potentiation of calcium deposition (Figure 5B) and an increase in TNAP activity (Figure 5C). Because protein kinase A (PKA) was shown to be involved in the vascular calcification process, in particular by directly activating TNAP, and is a second messenger of EET signalling [26,27,28,29], we tested the impact of the PKA inhibitor (PKI; 5–24) (10 µM). PKI alone did not modify calcium deposition or TNAP activity but prevented the effects of exogenous EETs (Figure 5B,C). We also confirmed the protective role of sEH hydrolase against the calcification of human VSMCs induced by EETs. Indeed, the increase in the calcium content induced by high-phosphate in VSMCs was not modified by exogenous EETs or *t*-AUCB added to the culture medium separately but it was enhanced when both agents were combined (Figure 5D).

### 2.5. Relationships between the mRNA Levels of sEH and Markers of the Osteochondrogenic Differentiation in Human Carotid Tissues

Endarterectomy samples obtained from 34 patients (Appendix A
Appendix A) were dissected in two fragments: the atherosclerotic calcified plaque and the distant macroscopically intact tissue. A global gene expression profiling analysis revealed that the sEH mRNA level was decreased by 24% in human atherosclerotic calcified plaques compared to distant intact tissues (Figure 6A). The SMMHC mRNA level was reduced by 42% in the plaque while Runx2 was increased by 40% without changes in the Sox9 and Msx2 mRNA levels (Figure 6A). In addition, the mRNA level of sEH was positively correlated with SMMHC and negatively correlated with Runx2 mRNA levels (Figure 6B).

## 3. Discussion

The major finding of the present study is that sEH exerts a protective action against the development of vascular calcification by metabolizing EETs.

To unravel the role of sEH in the vascular calcification process, we submitted rat aortic rings to high-phosphate conditions in the absence and in the presence of the specific inhibitor *t*-AUCB and of exogenous EETs. The major advantage of *t*-AUCB towards sEH genetic invalidation is to specifically inhibit the hydrolase activity of sEH without affecting its second enzymatic activity, which is a phosphatase [21,30]. Quantification of calcium deposition and histological staining confirmed that 3.8 mM Pi and a culture duration of 7 days is needed to induce significant aortic calcification [16]. The decrease in SMMHC mRNA expression level and the increase in Runx2, Msx2 and Sox9 levels confirmed the transition of vascular cells toward an osteochondrogenic phenotype. In addition, there was an increased TNAP activity and decrease in the levels of the inhibitor of calcification PPi, as previously shown in aortas isolated from uremic rats [14]. This pathway is involved in mineralization deposition facilitating the formation of the final product of calcium phosphate salt reaction hydroxyapatite, but it also may directly promote the transition of VSMCs to a osteochondrogenic phenotype [31,32].

In this context, although high-phosphate did not significantly modify the aortic mRNA expression or metabolizing activity of sEH, we observed significant effects of sEH pharmacological inhibition on aortic calcification. Thus, *t*-AUCB further increased the calcification of rat aortic rings when added to high-phosphate ex vivo, demonstrating that sEH attenuates vascular calcification. Although *t*-AUCB was shown to inhibit sEH in the low nanomolar range in vitro [21], there was a concentration-dependent potentiation of calcification with a significant effect only observed at 10 µM, which may be related to the degradation of the drug in the culture medium or to a reduced diffusion into the aortic tissue and cells to inhibit this cytosolic enzyme. At this concentration, *t*-AUCB also induced marked increases in the mRNA expression level of the osteochondrogenic marker Runx2 and in TNAP activity. The level of PPi did not further decrease in the culture medium of aortic rings treated with *t*-AUCB, but this could be due to the presence of PPi in the culture medium under basal conditions. In support of this hypothesis, the inhibition of CYP450 epoxygenases in addition to prevent all the deleterious effects of *t*-AUCB markedly increased PPi levels. Furthermore, we observed that *t*-AUCB did not display a procalcifying effect on deendothelialized aortic rings or on human VSMCs while the addition of 11,12-EET or 14,15-EET to the culture medium reinitiated the effect of *t*-AUCB. Interestingly, the procalcifying effects of exogenous EETs on VSMCs was only observed in the presence of *t*-AUCB while, in aortas, exogenous EETs alone are sufficient to promote vascular calcification. Thus, the potentiation of vascular calcification by exogenous EETs was obtained either in the absence or in the presence of DHETs. Although we cannot exclude that DHETs are also involved, this demonstrates that sEH plays a protective role against vascular calcification by metabolizing endogenous EETs synthesized by endothelial cells. Moreover, the inhibition of PKA abolished the increase in TNAP activity and the potentiation of calcification induced by exogenous EETs, showing the involvement of this signaling pathway as previously reported for EET-mediated regulation of VSMC proliferation [29].

Finally, the Runx2 expression level was markedly increased while sEH expression was decreased in human carotid calcified plaques compared to adjacent lesion-free sites, and a striking inverse correlation was found between both expression levels, supporting our ex vivo findings on rat aortas about the protective role of sEH against the development of vascular calcification. Overall, our findings are in accordance with previous clinical studies showing that subjects carrying the sEH genetic polymorphism R287Q, known to decrease its activity, are at higher risk of developing coronary artery calcification [11]. Regarding the study suggesting that EETs may prevent aortic calcification in patients with primary aldosteronism, it should be noticed that only plasma DHETs levels have been measured and that conclusion of this work was only based on the assumption that DHET and EET levels are inversely correlated [12]. This may be incorrect because, as we previously showed in humans [20], the more EET production increases, the more DHET level is elevated and, in this case, the conclusion should have confirmed a deleterious role of EETs in vascular calcification.

Altogether, our results show that sEH prevents the development of vascular calcification by metabolizing EETs and by preventing a PKA-dependent increase in TNAP activity. Importantly, this work points out a potential side-effect of the inhibitors of sEH, which is currently under active development to treat cardiometabolic diseases [8,9], and the significance of these findings will have to be carefully assessed in humans in vivo. Because the results were obtained using in vitro and ex vivo models, additional experiments are needed to evaluate the in vivo significance of our finding. In addition, the relative contribution of EETs toward the possible role of DHETs and those of other sEH substrates and metabolites on the vascular calcification process remain to be assessed. Finally, because the phosphatase activity of sEH is involved in the metabolism of lysophosphatidic acids [33], which are lipid mediators also known to promote calcification [34], further experiments are warranted to determine whether sEH plays a dual role in the regulation of vascular calcification.

## 4. Material and Methods

An expanded detailed description of the Methods section is available in the Online Appendix A.

### 4.1. Animal Experiments

Protocols for rat experiments were approved by the local institutional review committees (number 01353.01, 12 January 2015) and conducted in accordance with the Directive 2010/63/EU of the European Parliament. For ex vivo mineralization assays, 10-weeks-old male Wistar rats (Janvier Lab, Le Genest-Saint-Isle, France) were euthanized by intraperitoneal injection of pentobarbital (120 mg/kg), and the thoracic and abdominal aortas were dissected as previously described [14,15]. After removal of fat and connective tissues, the vessels were cut into 2- to 3-mm rings and placed in regular medium (Dubelcco’s Modified Eagle’s Medium; DMEM 6546, Sigma-Aldrich, Saint-Louis, MI, USA) containing 0.9 mM inorganic phosphate (Pi) or in high-phosphate DMEM with 3.8 mM Pi supplemented with 10% fotal bovine serum (FBS) at 37 °C in 5% CO_2_ for 7 consecutive days. The medium was changed at days 3 and 6.

### 4.2. Human Experiments

Human arterial tissues were obtained from patients undergoing aortic surgery or carotid endarterectomy who gave written informed consent, and studies were conducted in accordance with the Principles of Good Clinical Practice and the Declaration of Helsinki and were approved by the local ethical committee (Protocol number 2009-19, 7 August 2019).

Aortic VSMCs were isolated from aortic lesion-free explants, as previously described and validated [15,23,24]. For mineralization assays, human VSMC were cultured between P3 and P8 (100.000 cells/well; 6-well plates) in regular (0.9 mM Pi) or high-phosphate DMEM (3 mM Pi) supplemented with 1% FBS at 37 °C in 5% CO_2_ for 14 consecutive days. The medium was changed every 2 to 3 days.

Carotid endarterectomy samples from each patient were collected in the surgery room and immediately dissected in two fragments: the atherosclerotic calcified plaque and the distant macroscopically intact tissue. Each fragment was immediately frozen in liquid nitrogen and, intrapatient comparison of gene expression profile was performed using Affymetrix GeneChip Human Gene 1.0 ST arrays (Affymetrix, Santa Clara, CA, USA) as previously described [35].

### 4.3. Statistics

Statistics were performed using the SYSTAT package (SYSTAT 8.0; SPSS, Chicago, IL, USA). Results are presented as mean ± SEM. For ex vivo rat aortic and in vitro human VSMC mineralization assays, all experiments have been performed using at least 3 different aortic tissues from rats or humans. Analyses of the differences between two groups were performed by student paired t test or Wilcoxon matched paired test for nonnormality distributed data. Differences between 3 or more groups were analyzed using a generalized linear model with group as the factor and the rat or human aorta used as the cofactor, followed by Tukey–Kramer multiple comparison tests for post hoc analysis. Pearson correlation analyses were used to assess the relationships between human carotid mRNA expression levels of genes of interest. A value of *p* < 0.05 was considered statistically significant.

## Figures and Tables

**Figure 1 ijms-21-04313-f001:**
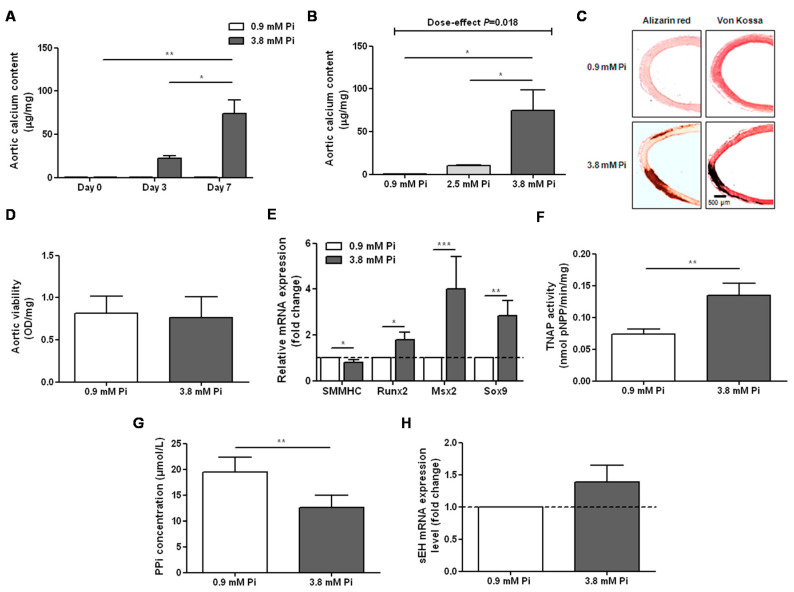
Time- (**A**) and dose- (**B**) dependent effect of inorganic phosphate (Pi) on the calcium content of aortic rings (*n* = 4–8 per condition) and representative images of Alizarin red and Von Kossa staining of aortic rings cultured for 7 days in 0.9 mM and 3.8 mM Pi (**C**): The impact of high-phosphate conditions on the viability of rat aortic rings was assessed using a methylthiazolyldiphenyl-tetrazolium assay ((**D**) *n* = 4 per group). Aortic mRNA expression levels of the contractile marker smooth muscle myosin heavy chain (SMMHC) and of the osteochondrogenic markers runt-related transcription factor 2 (Runx2), Msh homeobox 2 (Msx2) and sex-determining region Y-box 9 (Sox9) after 7 days of culture in normal (0.9 mM Pi) and high-phosphate (3.8 mM Pi) conditions ((**E**) *n* = 10–24 per group). Tissue-nonspecific alkaline phosphatase (TNAP) activity (**F**) assessed by measuring the hydrolysis of p-nitrophenyl phosphate (pNPP) and pyrophosphate anions (PPi) level (**G**) in the supernatants of aortic rings cultured for 7 days in 0.9 mM and 3.8 mM Pi (*n* = 14 per group). Aortic mRNA expression levels of soluble epoxide hydrolase (sEH) after 7 days of culture in 0.9 mM and 3.8 mM Pi (**H**). * *p* < 0.05, ** *p* < 0.01 and *** *p* < 0.001.

**Figure 2 ijms-21-04313-f002:**
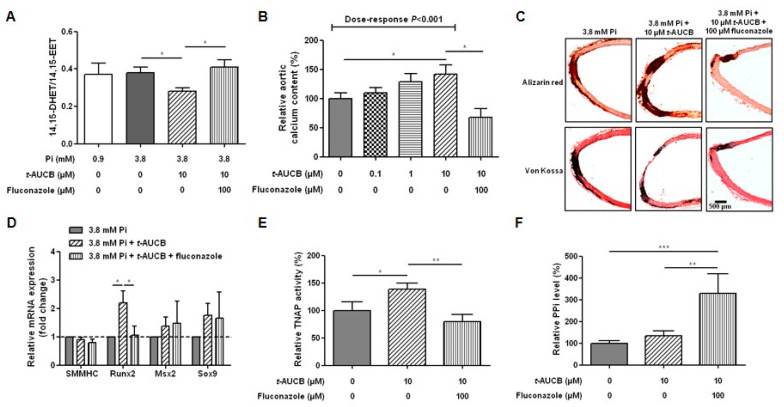
Ratio of 14,15-epoxyeicosatrienoic acid-to-14,15-dihydroxyeicosatrienoic acid (14,15-EET-to-14,15-DHET) in the supernatants of aortic rings cultured for 7 days in normal (0.9 mM inorganic phosphate, Pi) and high-phosphate (3.8 mM Pi) conditions in the absence and in the presence of 10 µM *trans*-4-(4-(3-adamantan-1-yl-ureido)-cyclohexyloxy)-benzoic acid (*t*-AUCB) and in 10 µM *t*-AUCB + 100 µM fluconazole ((**A**) *n* = 4–9 per group). The relative calcium content of aortic rings cultured for 7 days in 3.8 mM Pi in the absence and in the presence of increasing concentrations of *t*-AUCB alone and associated with fluconazole ((**B**) *n* = 6–16 per group) and representative images of Alizarin red and Von Kossa staining (**C**). Aortic mRNA expression levels of the contractile marker smooth muscle myosin heavy chain (SMMHC) and of the osteochondrogenic markers Msh homeobox 2 (Msx2), sex-determining region Y-box 9 (Sox9) and runt-related transcription factor 2 (Runx2) ((**D**) *n*= 4–22 per group) tissue-nonspecific alkaline phosphatase (TNAP) activity ((**E**) *n* = 13–18 per group) assessed by measuring the hydrolysis of p-nitrophenyl phosphate (pNPP) and pyrophosphate anions (PPi) level ((**F**) *n* = 4–13 per group) in culture supernatants after 7 days of culture in high-phosphate (3.8 mM Pi) conditions in the absence and in the presence of 10 µM *t*-AUCB and 10 µM *t*-AUCB + 100 µM fluconazole. * *p* < 0.05, ** *p* < 0.01 and *** *p* < 0.001.

**Figure 3 ijms-21-04313-f003:**
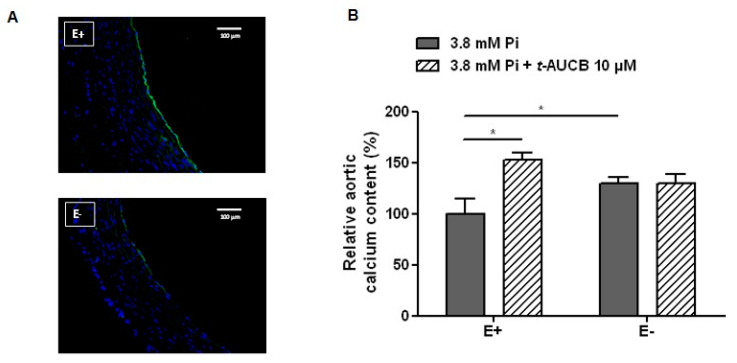
Representative images of CD31 immunostaining (**A**) and relative calcium content of intact (E+) and deendothelialized (E−) aortic rings cultured for 7 days in 3.8 mM Pi in the absence and in the presence of 10 µM *t*-AUCB ((**B**) *n* = 3–8 per group). * *p* < 0.05.

**Figure 4 ijms-21-04313-f004:**
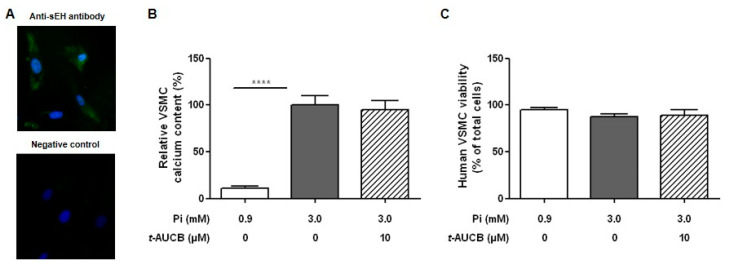
Representative images of sEH immunostaining in human vascular smooth muscle cells (VSMCs) (**A**): Relative calcium content ((**B**) *n* = 5–9 per group) and trypan blue viability assay ((**C**) *n* = 7–13 per group) of human VSMCs cultured in normal (0.9 mM inorganic phosphate, Pi) and high-phosphate (3.0 mM Pi) conditions for 14 days in the absence and in the presence of 10 µM *t*-AUCB. **** *p* < 0.0001.

**Figure 5 ijms-21-04313-f005:**
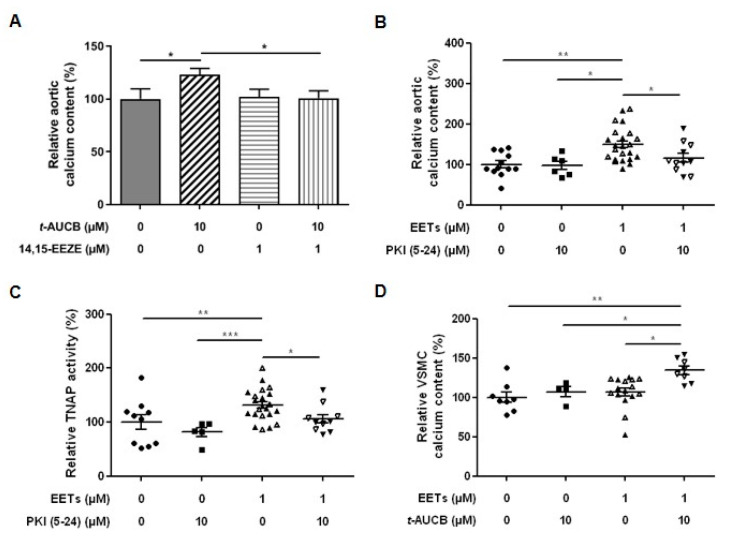
Relative aortic calcium content after 7 days of culture in high-phosphate conditions (3.8 mM inorganic phosphate, Pi) in the absence and in the presence of 10 µM *t*-AUCB and the EET antagonist 14,15-epoxyeicosa-5(Z)-enoic acid (14,15-EEZE) at 1 µM ((**A**) *n* = 6 per group). Relative aortic calcium content (**B**) and tissue-nonspecific alkaline phosphatase (TNAP) activity (**C**) in culture supernatants after 7 days of culture in 3.8 mM Pi in the absence and in the presence of 11,12-EET and 14,15-EET at 1 µM and the protein kinase A inhibitor (PKI; 5-24) at 10 µM. Relative calcium content of human VSMCs cultured in normal (0.9 mM Pi) and high-phosphate (3.0 mM Pi) conditions for 14 days in the absence and in the presence of 11,12-EET and 14,15-EET at 1 µM and 10 µM *t*-AUCB (**D**). Data are presented as mean ± SEM with individual values. Results obtained with 11,12-EET (solid symbols) and 14,15-EET (open symbols) have been pooled for statistical analysis. * *p* < 0.05, ** *p* < 0.01 and *** *p* < 0.001.

**Figure 6 ijms-21-04313-f006:**
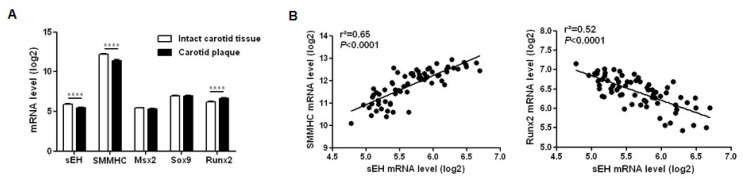
(**A**) mRNA expression levels of soluble epoxide hydrolase (sEH); of the contractile marker smooth muscle myosin heavy chain (SMMHC); and of the osteochondrogenic markers Msh homeobox 2 (Msx2), sex-determining region Y-box 9 (Sox9) and runt-related transcription factor 2 (Runx2) in calcified carotid plaques (*n* = 34) and distant intact tissues (*n* = 34). (**B**) relationships between sEH and SMMHC or Runx2 mRNA levels. **** *p* < 0.0001.

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
