# Peer review of "The Metabolism of Epoxyeicosatrienoic Acids by Soluble Epoxide Hydrolase Is Protective against the Development of Vascular Calcification"

_ijms, 2020, doi:10.3390/ijms21124313_

Round 1

Reviewer 1 Report

The authors have done an excellent job replying to previous comments and added new experiments.  

Reviewer 2 Report

None

This manuscript is a resubmission of an earlier submission. The following is a list of the peer review reports and author responses from that submission.

Round 1

Reviewer 1 Report

In the current manuscript the authors present data demonstrating the protective role of sEH against the development of vascular calcification.  Using a combination of in vitro approaches they suggest sEH hydrolase protects the vascular by reducing the levels of EETs.  This is a an interesting study that presents novel data, which is in contrast to some published views regarding sEH. 

Comments:

1)  The authors suggest EETs are the primary mediators of the adverse effects.  Further supportive data using the putative antagonist 14,15-EEZE and the diol metabolites such as 14,15-DHET is required.

2)  The authors utilize an in vitro model of vascular calcification, however, this injury is over a short time course and a single concentration.  Some discussion to support/rationalize the use and applicability of the model is required.

3) The inclusion of human tissue is a strength.  However, more discussion about the samples is required (ie., age, sex, etc).  In addition, the figure indicates n=34, does that mean 34 different humans or were multiple sections assessed from the same samples?  Importantly, the reported differences in sEH mRNA (Figure 6A), while statistically significant does not appear to be biological real. The difference in expression in the figure appears to be negligible and makes it hard to believe the conclusions. More evidence such as protein expression, catalytic activity and expanded metabolite profile is required.

4) Supporting evidence using a different approach to inhibit sEH would strengthen the data.

5) The manuscript would benefit from some editing.

Author Response

Response to reviewer 1

We thank the reviewer for his helpful comments.

1)  “The authors suggest EETs are the primary mediators of the adverse effects.  Further supportive data using the putative antagonist 14,15-EEZE and the diol metabolites such as 14,15-DHET is required.”

As suggested by the reviewer, the potentiation of aortic calcification by sEH inhibition and the prevention of this effect with CYP450 inhibition do not directly demonstrate that EETs are the primary mediators. Indeed, sEH metabolizes EETs to DHETs and the results obtained could have suggested that DHETs play a protective effect against vascular calcification, a protection alleviated when blocking sEH. Moreover, sEH metabolizes other epoxides formed from the action of CYP450 on linolenic acid and omega-3 fatty acids to a variety of diols, and all of these factors could also play a role in vascular calcification. Although of potential interest, we did not use 14,15-EEZE to go further because this agent blocks a not yet identified EET receptor, could be not effective in blocking other cellular targets of EETs and exhibits agonistic activity in some tissues (Harrington LS et al, Eur J Pharmacol 2004; Liu X et al, Prostaglandins Other Lipid Mediat 2017). Thus, we directly evaluated the impact of EETs on both the aortic model and human VSMC and showed a procalcifying effect. This was observed either in absence or presence of t-AUCB and thus in absence and presence of DHETs, clearly demonstrating that EETs potentiate calcification. This does not exclude that beyond EETs, DHETs and other sEH substrates and metabolites also play a role in vascular calcification and further experiments are needed to specifically address this hypothesis. To take into consideration this point, we added in the discussion that the potentiation of vascular calcification by exogenous EETs was obtained either in absence or in presence of DHETs and that although we cannot exclude that DHETs are also involved, this suggests that sEH plays a protective role against vascular calcification by metabolizing endogenous EETs synthesized by endothelial cells (lines 223-226). In addition, we now specified in the perspectives that it remains to be assessed the relative contribution of EETs toward the possible role of DHETs and those of other sEH substrates and metabolites on the vascular calcification process (lines 248-249).

2)  “The authors utilize an in vitro model of vascular calcification, however, this injury is over a short time course and a single concentration.  Some discussion to support/rationalize the use and applicability of the model is required.”

We agree with the reviewer that the development of vascular calcification relies on the interaction between multiple biological factors over a long period of time. Although the in vitro and ex vivo models probably only partially recapitulate the pathophysiology of the disease, they have been used by our group and many others to assess the putative mechanisms involved and impact of therapies in good time. These models are usually based on the exposure of blood vessels or cells to uremic toxins and in particular to inorganic phosphate to mimic the chronic kidney disease state that is the main risk factor associated with vascular calcification. For this work, we first revalidated the aortic model showing that a dose of 3.8 mM inorganic phosphate during 7 days is needed to induce significant calcium deposition and osteochondrogenic differentiation (RT-PCR experiments) and increased alkaline phosphatase activity (Figure 1). In addition, for the studies on human VSMC we used 3 mM inorganic phosphate during 14 days based on a previous validation study by our group (Louvet L et al, Nephrol Dial Transplant, 2013; reference 23). To take into consideration this point, we now specified in the results (lines 104) and methods (lines 275) sections that the in vitro model has been previously validated. We already described the validation of our aortic model at the beginning of the discussion section (lines 196-201) but we now specified in the perspectives that, because the results were obtained using in vitro and ex vivo models, additional experiments are needed to evaluate the in vivo significance of our finding (lines 246-248). In addition, we toned down our conclusions (lines 208, 224-225, 241).

3) “The inclusion of human tissue is a strength.  However, more discussion about the samples is required (ie., age, sex, etc).  In addition, the figure indicates n=34, does that mean 34 different humans or were multiple sections assessed from the same samples?  Importantly, the reported differences in sEH mRNA (Figure 6A), while statistically significant does not appear to be biological real. The difference in expression in the figure appears to be negligible and makes it hard to believe the conclusions. More evidence such as protein expression, catalytic activity and expanded metabolite profile is required.”

As requested by the reviewer, we now described the clinical characteristics of the patients who underwent endarterectomy in the supplemental Table 1. Thirty-four patients were included in the study and the carotid endarterectomy samples were immediately dissected in two fragments: the atherosclerotic calcified plaque and the distant macroscopically intact tissue, allowing intrapatient comparison of the transcript profiles. This is now better described in the results (lines 163-164) and methods section (lines 278-282). The mRNA expression was expressed as log2 in Figure 6 and we agree with the reviewer that this appears to underestimate the true difference between carotid plaques and adjacent sites. In fact, sEH mRNA expression was reduced by 24% in carotid plaque, SMMHC expression by more than 42% while Runx2 expression was increased by 40%. These variations were now described in the results sections (lines 164-168), strengthening our conclusions. Although the determination of sEH protein expression and activity would have been of interest, both fragments of endarterectomy samples fragment were already divided: a part was used for RNA analysis, whereas the other was used for histological examination, and no more tissue was available for further analysis.

4) “Supporting evidence using a different approach to inhibit sEH would strengthen the data.”

We agree with the reviewer that a different approach to inhibit sEH may have been useful to strengthen our results. This could have been done using aortas from sEH knockout mice or siRNA. However, sEH is a bifunctional enzyme also possessing a phosphatase activity whose biological function remains unknown and using these approaches lead to block both the hydrolase and phosphatase activities. In fact, previous studies have shown that the genetic invalidation of sEH did not always lead to the same effects than those obtained when blocking sEH with highly specific pharmacological inhibitors that did not affect the phosphatase activity such as t-AUCB. Nonetheless, to take into consideration this point, we now specified in the discussion section that the major advantage of t-AUCB towards sEH genetic invalidation is to specifically inhibit the hydrolase activity of sEH without affecting its second enzymatic activity, which is a phosphatase (lines 194-196). In addition, we also added in the perspectives that because the phosphatase activity of sEH is involved in the metabolism of lysophosphatidic acids, which are lipid mediators also known to promote calcification, further experiments are warranted to determine whether sEH plays a dual role in the regulation of vascular calcification (lines 250-252). References have been updated accordingly (references 29, 32 and 33.

5) The manuscript would benefit from some editing.

As requested by the reviewer, typos and grammatical errors have been corrected by the contributing author from the University of Davis, USA, Dr C. Morisseau, and can be viewed in the tracked mode version.

Reviewer 2 Report

The current study demonstrates that high phosphate induced aortic calcification is enhanced by the soluble epoxide hydrolase inhibitor t-AUCB. Rat aortic rings exposed to high phosphate demonstrate increased calcification in response to t-AUCB. Additional experiments demonstrated the contribution for EETs and Runx2 expression. These data were coupled with data from human carotid calcified plaques demonstrating decreased sEH expression and increased Runx2 expression. Overall, this is an interesting study providing novel findings on EETs and soluble epoxide hydrolase.

"Data not shown" is not appropriate. The data should be provided in the manuscript or supplementary materials. It is interesting that it required a high dose for t-AUCB to increase aortic calcification. This is an inhibitor with a nanomolar IC50. The lack of effect with 0.1 and 1 uM t-AUCB needs to be addressed in the discussion.

Author Response

Response to reviewer 2

We thank the reviewer for his comments.

Point 1. "Data not shown" is not appropriate. The data should be provided in the manuscript or supplementary materials.

As requested by the reviewer, we now added the data showing the absence of effect of 10 µM t-AUCB on the calcium content of aortic rings cultured in 0.9 mM Pi in the Supplementary Figure 1.

Point 2. It is interesting that it required a high dose for t-AUCB to increase aortic calcification. This is an inhibitor with a nanomolar IC50. The lack of effect with 0.1 and 1 µM t-AUCB needs to be addressed in the discussion.

As suggested by the reviewer, we added some arguments in the discussion to explain why the procalcifying effect of t-AUCB was only significant at the highest concentration. We thus specified that, although t-AUCB was shown to inhibit sEH in the low nanomolar range in vitro, there was a concentration-dependent potentiation of calcification with a significant effect only observed at 10 µM, which may be related to the degradation of the drug in the culture medium or to a reduced diffusion into the aortic tissue and cells to inhibit this intracellular enzyme. Alternatively, besides its great selectivity for sEH, t-AUCB may target another protein at high concentration (lines 209-213). Importantly, the results obtained with exogenous EETs confirmed the role of sEH in vascular calcification.

Round 2

Reviewer 1 Report

In the revised version, the authors provide written responses to previous concerns but have not provided any new data.  There still remains a concern about key supportive data, notably experiments using 14,15-EEZE and DHET.  While the authors discuss different effects of 14,15-EEZE and suggest it cannot be used, there is strong evidence in literature demonstrating antagonizing properties.  In addition, concerns raised by the other reviewer highlighted the high concentration of tAUCB used in the experiments, which might suggest off target effects of the sEHi  - there seems to be some inconsistency with the authors rebuttal arguments between rationalizing the use of 14,15-EEZE and tAUCB. Importantly, as the authors are proposing the metabolism of EETs is main protective effect, more supportive data is required or they should consider altering and/or changing the 'title, hypothesis, conclusions, etc" to the better reflect the data such as something more like the protective effects of tAUCB would be appropriate.

Author Response

We thank the reviewer for his helpful comments.

As requested by the reviewer, we performed additional experiments using the EET antagonist 14,15-EEZE to provide supporting data showing that EET metabolism by sEH is protective against vascular calcification. Indeed, we observed that 14,15-EEZE alone did not change the calcium content of aortic rings cultured in high-phosphate conditions but prevented the increase induced by t-AUCB (Figure 5A). Accordingly, we specified in the results section that “because cytochrome P450 enzymes blocked by fluconazole not only produce EETs but also other epoxides from various omega-3 and omega-6 fatty acids, we performed additional independent experiments using the EET antagonist 14,15-epoxyeicosa-5(Z)-enoic acid (14,15-EEZE; 1 µM). Addition of 14,15-EEZE alone did not change the calcium content of aortic rings cultured in 3.8 mM Pi but prevented the increase induced by 10 µM t-AUCB“ (page 4, lines 153-157). These results are presented on Figure 5A and a reference supporting the use of 14,15-EEZE as EET antagonist has been added (reference 26). The abstract has also been modified according with these additional results: “The procalcifying effect of t-AUCB was prevented by mechanical aortic deendothelialization or inhibition of the production and action of epoxyeicosatrienoic acids using the cytochrome P450 inhibitor fluconazole and the antagonist 14,15-EEZE respectively”.
